# Explosive Spalling Mechanism and Modeling of Concrete Lining Exposed to Fire

**DOI:** 10.3390/ma15093131

**Published:** 2022-04-26

**Authors:** Rujia Qiao, Yinbo Guo, Hang Zhou, Huihui Xi

**Affiliations:** 1School of Science, Xi’an University of Architecture & Technology, Xi’an 710055, China; grenbo_yyds@163.com; 2School of Civil Engineering, Xi’an University of Architecture & Technology, Xi’an 710055, China; zhouhang@xauat.edu.cn (H.Z.); xihuihui1996@163.com (H.X.)

**Keywords:** explosive spalling, concrete lining, tunnel fire, multilayer model, spalling depth prediction

## Abstract

Traditional heat transfer analysis has been adopted to predict the damage in a tunnel under fire without considering the effect of concrete spalling, which leads to underestimation of the fire damage of concrete. However, accounting for the spalling effect of concrete under high temperature in an analytical heat transfer model is difficult because of the complexity of the spalling mechanism. This study aims to establish an analytical model to estimate the influence of concrete spalling on the fire-damage depth prediction. To overcome this challenge, first, a series of fire tests were conducted in a unidirectional heating system. The spalling phenomenon and spalling characteristics were observed. Based on the experimental test results, the moisture content of concrete is one of the key factors of spalling. Obvious layered spalling characteristics of concrete samples without drying could be observed under the unidirectional heat conduction system. The critical temperature of spalling is 600 °C, and the thickness of the spalling layer is 2 cm~2.5 cm. These two parameters are critical spalling conditions. Second, a multilayer model for the heat transfer analysis considering the spalling effect of tunnel lining under fire was proposed. By using Laplace transform and the series solving method for ordinary differential equations, the time-dependent temperature and stress fields of concrete lining during tunnel fire could be obtained, which are the basis of damage evolution. The analytical results agreed with the experimental data. The spalling depth of tunnel lining related to the temperature rise of tunnel fire could be predicted by using the proposed analytical model. The results of this research can be used to provide a better damage evaluation of tunnel lining under fire.

## 1. Introduction

Fire safety in tunnels and underground spaces is an important topic [1]. Besides fire alarms, evacuation, ventilation and smoke exhaustion, structural safety after fire is also an important issue [2]. Compared to building fires, tunnel fires with characteristics of high temperature, rapid heating rate, long duration and nonuniform temperature distribution inside the tunnel are more dangerous and destructive [3,4]. When tunnel fire temperature rises rapidly, one of the most serious factors of damage is the explosive spalling of the concrete lining, where the cracked or delaminated surface layer of the concrete peels in a progressive or explosive way. The spalling of the concrete lining and property degradation will reduce the thickness of the lining section and generate instability [5,6,7]. Therefore, it is very important to quantitatively predict the damage and structural loss of tunnel lining caused by fire so as to achieve the best fire protection design and recovery after fire.

The prediction of fire damage of structures has been studied based on conventional heat transfer analysis in most research [8,9,10]. However, conventional heat transfer analysis is acceptable only when no spalling has occurred. The risk of fire-induced damage in a tunnel can be underestimated if the structural loss induced by spalling is neglected. Experimental tests can evaluate fire-induced structural damage and its damage depth, but it is time-consuming, costly and limited by sample type [11,12,13]. As reported by Choi et al., a new heat transfer analysis model considering the spalling of concrete under high temperature is needed to accurately and reasonably predict the fire damage of tunnel lining [14].

To establish such a new heat transfer analysis model, the spalling characteristics and spalling mechanism of concrete lining exposed to fire are key considerations. The spalling phenomenon of ordinary concrete under high temperature was first reported by Harmathy in 1965 [15]. Concrete is prone to spalling when the temperature between 200 °C and 500 °C. There are many factors that affect the spalling of concrete. Internal factors include the water–cement ratio, moisture content, aggregate types, permeability, etc. [16,17,18]. External factors include the heating rate, fire temperature, environmental conditions (temperature and humidity), load level, etc. [19,20,21,22]. The heating rate is one of the important factors [23]. The faster the heating rate, the higher the probability of spalling. Pore pressure rise induced by water vaporization is another important cause of concrete spalling [24,25,26,27]. Moisture content, permeability and heating rate are the main parameters affecting pore pressure [28]. For the critical spalling condition, Tenchev et al. proposed that 300 °C is the critical temperature at which concrete starts to become damaged [29]. Khoury et al. presented that the heated surface of ordinary concrete will crack or spall when the temperature rises to 250~420 °C [12]. To date, it is still not entirely clear which factor is responsible for thermal spalling. Therefore, the objective of this paper is to investigate the spalling characteristics and spalling mechanisms of concrete under tunnel fire.

According to the driving mechanism, the spalling of concrete has certain regularity on the macro scale. The temperature and pore pressure in concrete change in a gradient and have a clear layering phenomenon. The layering phenomenon is more obvious in tunnel lining due to the unidirectional heat conduction of tunnel fire. Thus, using a multilayer model is a rational and effective approach, especially for reinforced concrete [30,31]. Yan et al. presented a multilayer thermoelastic damage model to analyze the bending behavior of a tunnel lining segment exposed to fire [32]. However, fire-induced spalling was not considered in their research. In order to assess the stability of shallow tunnels, Savov et al. used layered finite beam elements to discretize the tunnel lining and considered spalling by deactivating the layer following a critical condition [33]. In their analysis, displacement and stress could be obtained under a preset spalling thickness. The thickness of the spalling layer is a hypothetical value and not obtained experimentally. Cheng et al. developed an efficient numerical model to reproduce the fire-induced spalling process taking place in concrete and to investigate the underlying mechanism of thermal spalling [34]. The research is conducted from the perspective of crack propagation. Choi et al. developed a finite-element model that was able to simulate the spalling phenomenon during fire by eliminating elements exceeding a predetermined critical temperature in the analysis [35]. The simulation results were much closer to the experimental results when the thickness of the spalling layer was 2.5 cm and the critical temperature of the spalling section was 600 °C. However, the numerical simulation results were greatly affected by the size of the element. To sum up, the explosive spalling mechanisms of concrete still have no unified conclusion. Few studies have established an analytical model to estimate the influence of concrete spalling on the fire-damage depth prediction.

In this study, a multilayer analytical model considering the spalling effect of concrete lining under fire is established according to the fire test results. This has not appeared in previous studies. This paper is organized as follows: In Section 2, the sample preparation and experiment setup of the tunnel fire test are described; in Section 3, the characteristics of delamination spalling are analyzed, the spalling thickness and temperature are discussed and the critical failure conditions are given; in Section 4, an analytical model is established based on the experimental results; in Section 5, temperature and stress distribution of tunnel lining and spalling depth are discussed; finally, concluding remarks are drawn. It is verified that the calculated results of the theoretical model are in good agreement with the experimental results. This analytical model can be used to analyze the influence of concrete spalling on the prediction of damage depth under different fire scenarios.

## 2. Experimental Procedure

### 2.1. Preparation of Concrete Samples

Concrete samples were prepared for fire tests, with dimensions of 300 mm × 100 mm × 100 mm designed. The size details of these samples are shown in Figure 1. The concrete samples were employed to represent ordinary secondary concrete lining, with a design compressive strength of 50 MPa at an age of 28 days. All the samples were demolded after 1 day of pouring. After 28 days of standard curing at a temperature of 26 °C, the preparation and curing of specimens were carried out according to the requirements of the Standard for Test Methods of Concrete Physical and Mechanical Properties (GB/T 50081-2019). The samples were left to age and dry in a laboratory for one year. Before the fire tests, the average moisture content of the samples was measured. The moisture content of the concrete samples *ρ_i_* can be expressed as follows:*ρ_i_* = (*m_i_* − *m*_0_)/*m*_0_(1)
where *m_i_* is the weight of the concrete samples that were left to age and dry in a laboratory for one year. *m*_0_ is the weight of the concrete samples dried for 36 h.

Because long-term aging in normal conditions cannot significantly affect the weight and water content of specimens, the average moisture content was employed in our experiments to describe the moisture content of concrete. The average moisture content was 2.1%, which was the average value of the moisture content of 10 samples in the same batch. The mixture proportions of concrete are listed in Table 1. Concrete C40P8 was selected as the secondary tunnel lining. P8 indicates that the expansive agent was added to the cement in the proportion of 8%. This can compensate for the shrinkage of concrete and greatly improve the antiseepage and crack resistance with less cost. Concrete admixtures are substances added in the process of mixing concrete to improve its performance. Their amount is generally no more than 5% of the quality of cement.

As shown in Figure 2, four sides and the right end face of a ceramic fiber sheet were directly wrapped to ensure one-sided heating during the fire test. The ceramic fiber sheet can withstand temperatures as high as 1600 °C, with a thickness of 40 mm. Two layers of insulation covered the samples in the tests. Therefore, the actual thickness of the insulation was 80 mm. Using a ceramic fiber sheet can not only insulate heat conduction but also prevent the evaporation loss of water.

During the tests, five sections were measured, as shown in Figure 2. Five monitoring holes were created with an interval of 5 cm, a diameter of 4 cm and a depth of 5 cm. A K-type thermocouple was used to measure the temperature. Aluminum powder with high heat conductivity was poured into the monitoring holes to ensure that the thermocouple and concrete were in close contact. The maximum operating temperature and accuracy of the sheathed thermocouple were 1300 °C and ±0.75%, respectively.

### 2.2. Fire Test

The fire test was carried out in an automatic temperature-controlled muffle furnace (LYL-17LB) in the research, which was manufactured by LUOYANG LIYU KILN Co., Ltd. in Luoyang, China. The temperature in the muffle was automatically controlled by two programmable logic controllers (PLCs), with a maximum operating temperature of 1200 °C and temperature accuracy of ±0.1%, as shown in Figure 3.

A circular observation window was installed on the furnace door to allow observing the real-time changes of the samples in the muffle. The observation window was cooled by an automatic water circulation system. A wire hole was created on the back wall of the heating chamber to ensure the real-time monitoring of the internal temperature of the specimens during the heating process.

The left wall of the heating chamber was the heating zone (Ⅰ). The upper wall, the lower wall and the right wall constitute the heating zone (Ⅱ). When only the heating zone (Ⅰ) works, one-sided heating can be realized. When the heating zone (Ⅰ) and (Ⅱ) work together, uniform heating can be realized. The heating zone (Ⅰ) and (Ⅱ) are controlled by the PLC, which can set the temperature–time function as required. The concrete lining under tunnel fire has the characteristics of unilateral heat transfer. In order to ensure unilateral heating, only the left heating wall (Ⅰ) was processed after the muffle furnace was preheated.

The numbers in each case are listed in Table 2. The parameters of the heating curve are shown in Table 3. The heating rate was the same, i.e., 40 °C/min. The maximum temperatures of the fire tests were 800 °C and 1000 °C. The duration of the maximum temperature was 30 min. The natural cooling process was selected. During the test, the monitoring data from the thermocouples were recorded at 1 s intervals for each channel. In this study, the fire-induced cracking and spalling mechanism of the concrete samples in the unidirectional heat transfer fire test system were tested and analyzed.

## 3. Experimental Results

### 3.1. Spalling Phenomenon

The temperature inside the concrete samples was monitored during the tests. Figure 4 shows the temperature development inside the concrete samples. There was no spalling during the tests. The maximum temperatures of the heating curves were 800 °C and 1000 °C. In Figure 4a,c, the concrete samples were heated without drying. In Figure 4b,d, the concrete samples were dried at 105 °C for 36 h before the heating tests. The temperature curves of the section 5 cm from the heated surface had a platform stage when its temperature rose to about 100 °C in Figure 4a,c. The “platform stage” was the result of water vaporization and migration in the concrete. Meanwhile, the temperature curves of the section 5 cm from the heated surface were smooth in Figure 4b,d. This was because the samples were dried before the tests. The existence of the monitoring hole provides an escape channel for the vapor, which reduces the pore pressure and heat conversion inside the concrete. This may be the main reason for no spalling observed during the tests. This also illustrates that the water in the concrete had a great impact on its internal temperature development.

From the results in Figure 4a–d, it can be seen that the closer to the heated surface, the higher the temperature. For the case of the maximum fire temperature of 800 °C, the temperature in the section with 5 cm from the heated surface was only 330 °C, and the temperature was 167 °C in the section with 10 cm from the heated surface. This implies that the temperature decayed rapidly inside the concrete. Once the distance from the heated surface was more than 15 cm, the temperature was less than 200 °C, which had little effect on the performance degradation of concrete. The influence depth of fire was shallow when no spalling occurred.

Figure 5 shows the four sides of the concrete samples after the heating tests. There are some cracks near the monitoring holes. In turn, only one crack can be seen on the heated surface. This means that the crack first initiated near the monitoring hole and then propagated to the heated surface. The number of cracks in Figure 5a is significantly more than in Figure 5b. This is because the concrete samples used in Figure 5b were dried, while the samples used in Figure 5a were not. Based on the preceding analyses, it can be concluded that the water in the concrete and the existence of the monitoring hole had a significant impact on its internal temperature development, even on spalling. It is very important and necessary to consider the influence of the moisture content and the setting of the monitoring hole in concrete fire tests.

In the previous tests, no spalling phenomenon of concrete was found when the temperature inside the concrete samples was monitored. However, this indirectly proves the influence of the existence of water in concrete on its spalling failure. In order to further verify the impact of the water content on the spalling of concrete, the samples were heated directly without setting internal monitoring holes. This can avoid the overflow of water vapor through the temperature monitoring holes and reduce the internal steam pressure. During the fire tests, a burst sound was suddenly heard when the temperature of the heated surface reached about 625 °C (mean value) for the test with the maximum fire temperature of 800 °C. For the case of the maximum temperature of 1000 °C, the burst sound was suddenly heard when the heated surface temperature was about 614 °C (mean value). Limited by the test conditions, the debris from the spalling was scattered between the heat source and the new heated surface, which reduced the heat transfer of the new heated surface. Thus, no secondary spalling occurred in the experiments. At the end of the fire tests, it was found that the surface layer fell off completely from the sample when the furnace door opened. The inner wall of the muffle chamber was destroyed. This implies that the spalling was violent.

Figure 6 shows the results of the fire tests without monitoring the internal temperature of samples. The four sides of the concrete samples were presented after the fire tests with maximum temperatures of 1000 °C and 800 °C. The spalling phenomenon was observed in both cases, and the whole layer of spalling could be observed. The layered characteristic of the concrete after the fire tests can be clearly seen in Figure 6a,b. Along the direction of heat conduction, the first layer is the spalling layer, which is grayish-white. The second layer is the drying layer, which is yellowish-brown. The third layer is the moisture gain layer, which is obviously blackened. The remaining part can be set as the fourth layer, which has no obvious change. The color change of concrete is consistent with the temperature gradient of concrete under unidirectional heat transfer [36,37]. Compared with Figure 5, the layered phenomenon is more obvious, especially in the third layer. The water vapor migrated due to high temperature and accumulated when meeting the concrete with a low temperature in the process. The porosity, moisture content of concrete and heating rate have an important influence on this process.

The morphology, structure and composition of the fracture surface are illustrated in Figure 7. It can be seen that the breaking failure at high temperature includes both transgranular fracture and intergranular fracture. In Figure 7, the transgranular fracture and intergranular fracture in part of the fracture surface are marked in detail, where the red solid lines point to transgranular fracture and the yellow dotted lines to intergranular fracture. Intergranular fracture is obviously more than transgranular fracture. In addition, intergranular fracture is more easily formed for coarse aggregate.

The spalling part broke into five parts and some fragments. Visible cracks were observed on the spalling section, while no corresponding crack could be found on the residual section. This implies that the cracks caused by high temperature are parallel to the heated surface. With the increase in the temperature gradient and pore pressure, cracks propagate and penetrate. Finally, the whole layer separates from the sample suddenly. These visible cracks on the spalling section may be generated at the moment of spalling, or they may be caused by the spalling layer hitting the inner wall of the heating chamber.

According to the experimental phenomena and results, the spalling characteristics can be summarized as follows: (1) spalling suddenly occurred, and the first spalling occurred before the fire temperature reached the maximum value; (2) the surface layer was completely separated from the sample; (3) the temperature of the heated surface was just over 600 °C at the spalling moment in this study. The thickness of the spalling layers was 2 cm~2.5 cm. This phenomenon is consistent with Choi’s research. In this research, the simulation results were much closer to the experimental results when the thickness of the spalling layer was 2.5 cm and the critical temperature of the spalling section was 600 °C.

### 3.2. Spalling Mechanism

According to the results of the experiments, it can be concluded that water vapor migration has an important influence on the surface spalling of concrete under high temperature. Concrete is a porous composite medium. Under high temperature, the water inside the concrete vaporizes and forms pore pressure. The pore pressure difference is formed between the heated surface and the end surface, which promotes the movement of gas (air and vapor) in the concrete. When a lower temperature is encountered, the steam recondenses to form a “quasi-saturated layer” and continuously moves to the outside of the lining. This is one of the main driving forces of concrete spalling in a layer. Another main driving force is the thermal stress caused by the heat transfer. The temperature of the heated surface of the lining increases under fire. The heat is transferred from the heated surface to the end, resulting in the generation of a temperature field. The temperature transfer leads to the thermal expansion gradient and then produces the tensile stress perpendicular to the heating surface. Thermal stress and steam pressure finally lead to the generation of strain. Stress and strain are the direct factors causing high-temperature cracking of concrete.

According to the spalling characteristics, it is obvious that the spalling caused by one-sided heating could be regarded as a layering fracture failure. There are two very important parameters: the critical temperature of spalling and the thickness of the spalling layer. Their values are affected by the material parameters and the heating rate of fire. In this study, the critical values of spalling according to the experimental results are that the temperature of the heated surface reaches 600 °C, and the thickness of the spalling layer was 2 cm.

Here, we attempt to establish a multilayer model to investigate the thermomechanical behavior of tunnel lining under fire, where fire-induced spalling is considered. As shown in Figure 8, the inner boundary of the model becomes a moving boundary: the inner boundary of the model is at Position 1 at the beginning. The pore pressure and temperature gradient increase with the temperature. The first layer will peel off when it reaches the critical temperature. Then, the inner boundary of the model moves to Position 2. At this time, the thermophysical properties of the remaining part also change with the increase in temperature. As the temperature continues to rise, the second layer will peel off, and the inner boundary will move to Position 3. Likewise, the inner boundary of the model will continue to move.

## 4. Analytical Prediction of Layered Spalling of Tunnel Lining

A multilayer thermoelastic model is proposed to analyze the temperature and stresses of tunnel lining exposed to fire. The following assumptions are adopted to facilitate the development of the model: (1) the tunnel section is simplified as a circle; (2) concrete is isotropic at the macro level, and the damage caused by temperature is also isotropic at each layer; (3) the temperature distribution on the cross-section is uniform; (4) the material of each layer is assumed to be homogeneous and linearly elastic.

As shown in Figure 9, the inner and outer radii of the model are *r*_0_ and *r*_N_ in the cylindrical coordinates. The inner surface is the heated surface. *r_i_* is the outer radius of the *i*th layer. The superscript *i* represents the *i*th layer. *T_i_* is the outer boundary temperature of the *i*th layer. Based on the previous analysis, the *i*th layer will spall when T*_i−_*_1_ is over the critical temperature.

The effect of the convective heat transfer coefficient is considered at the inner and outer surfaces of the model. The heat transfer coefficients at the inner and outer surfaces are defined as *h*_1_ and *h*_2_. In addition, the external pressure *q_b_* applied to the tunnel lining is also considered in this paper.

The RABT fire curve is employed in this study to simulate the tunnel fire temperature. The outer surface temperature is determined according to the ambient temperature of the tunnel, which is defined as *T_b_*. Without loss of generality, the whole process of fire development can be divided into three stages; that is, temperature rising (0 *≤ t < t*_1_), temperature holding (*t*_1_
*≤ t < t*_2_) and cooling stages (*t*_2_
*≤ t < t*_3_). As shown in Figure 10, the maximum temperature is denoted as *T_a_*. The temperature achieves the maximum value *T_a_* at *t*_1_. The temperature starts to decrease at *t*_2_ and returns to the original temperature at *t*_3_.

In this paper, dimensionless parameters are introduced to simplify the calculation. See Appendix A for details.

(1)Governing equation of heat condition and boundary condition

Considering the dimensionless variables, the unsteady-state heat conduction equation for the *i*th layer can be expressed as:(2)$$\frac{\partial^{2}\Theta_{i}\left(R,\tau \right)}{\partial R^{2}}+\frac{1}{R}\frac{\partial \Theta_{i}\left(R,\tau \right)}{\partial R}=\frac{1}{K}\frac{\partial \Theta_{i}\left(R,\tau \right)}{\partial \tau },\;i=1,\;2,\;\ldots \;N$$
where Θ*_i_*(*R*, *τ*) is the dimensionless temperature in the *i*th layer, which is a function of *τ* (dimensionless time) and *R* (dimensionless radius). *K* is the dimensionless form of the thermal diffusion coefficient *k* (*k* = *λ*/*ρc*, within *λ*, *ρ*, *c* are heat conduction coefficients, density and specific heat capacity, respectively).

The RABT fire curve is employed in this study. It can be expressed as a piecewise function:(3)$$\Pi \left(\tau \right)=\{\begin{array}{cc} \frac{\Theta_{a}}{\tau_{1}}\tau & 0\leq \tau <\tau_{1}\\ \Theta_{a} & \tau_{1}\leq \tau <\tau_{2}\\ -\frac{\Theta_{a}}{\tau_{3}-\tau_{2}}\tau +\frac{\tau_{3}\Theta_{a}}{\tau_{3}-\tau_{2}} & \tau_{2}\leq \tau <\tau_{3} \end{array}$$
where Θ*_a_* is the dimensionless form of maximum temperature of fire. *τ*, *τ*_1_, *τ*_2_ and *τ*_3_ are the dimensionless forms of *t*, *t*_1_, *t*_2_ and *t*_3_.

The original temperature can be presented as:(4)$$\Theta_{i}(R,0)=0,i=1,2,\ldots N$$
(5)$$R=R_{0}:\frac{\partial \Theta_{1}\left(R_{0},\tau \right)}{\partial R}-H_{1}\Theta_{1}\left(R_{0},\tau \right)=-H_{1}\Pi (\tau )$$
(6)$$R=R_{N}:\frac{\partial \Theta_{N}\left(R_{N},\tau \right)}{\partial R}+H_{2}\Theta_{N}\left(R_{N},\tau \right)=H_{2}\Theta_{b}$$

The continuity conditions of temperature can be expressed as:(7)$$\Theta_{i-1}\left(R_{i-1},\tau \right)=\Theta_{i}\left(R_{i-1},\tau \right),i=1,2,\ldots N$$
(8)$$\Lambda_{i-1}\frac{\partial \Theta_{i-1}\left(R_{i-1},\tau \right)}{\partial R}=\Lambda_{i}\frac{\partial \Theta_{i}\left(R_{i-1},\tau \right)}{\partial R},i=1,2,\ldots N$$

The temperature of tunnel lining increases with exposure time. When *T*_0_ exceeds the critical temperature, the inner boundary *R*_0_ moves to *R*_1_ with the spalling of the first layer. The moving condition of the inner boundary can be simplified as:(9)$$\Theta_{1}\left(R_{0},\tau \right)\;\leq \Theta_{\mathrm{Critical}};\frac{\partial \Theta_{1}\left(R_{0},\tau \right)}{\partial R}-H_{1}\Theta_{1}\left(R_{0},\tau \right)=-H_{1}\Pi (\tau )$$
(10)$$\Theta_{1}\left(R_{0},\tau \right)>\Theta_{\mathrm{Critical}};\frac{\partial \Theta_{2}\left(R_{1},\tau \right)}{\partial R}-H_{1}\Theta_{2}\left(R_{1},\tau \right)=-H_{1}\Pi (\tau )$$
(11)$$\Theta_{2}\left(R_{1},\tau \right)>\Theta_{\mathrm{Critical}};\frac{\partial \Theta_{3}\left(R_{2},\tau \right)}{\partial R}-H_{1}\Theta_{3}\left(R_{2},\tau \right)=-H_{1}\Pi (\tau )$$
(12)$$\Theta_{i}\left(R_{i}{}_{-1},\tau \right)>\Theta_{\mathrm{Critical}};\frac{\partial \Theta_{i+1}\left(R_{i},\tau \right)}{\partial R}-H_{1}\Theta_{i+1}\left(R_{i},\tau \right)=-H_{1}\Pi \left(\tau \right)$$

In Equations (4)–(12), *R*_0_ and *R*_N_ are the dimensionless radii of the inner and outer surfaces of the model, respectively. *R_i_* is the dimensionless outer radius of the *i*th layer. Θ*_a_* and Θ*_b_* are the dimensionless temperatures of *T_a_* and *T_b_*, respectively. *H*_1_ and *H*_2_ are the dimensionless heat transfer coefficients of *h*_1_ and *h*_2_, respectively.
(2)Equilibrium equation and stress boundary conditions:
(13)$$\frac{d\Sigma_{r,i}\left(R_{i},\tau \right)}{dR}+\frac{\Sigma_{r,i}\left(R_{i},\tau \right)-\Sigma_{\theta ,i}\left(R_{i},\tau \right)}{R}=0$$where Σ*_r_*_,*i*_(*R**_i_*,*τ*) and Σ*_θ_*_,*i*_(*R**_i_*,*τ*) are the dimensionless radial stress and circumferential stress, respectively.

Introducing the geometric equation and stress–strain relationship into the equilibrium equation:(14)$$\frac{d^{2}U_{i}\left(R,\tau \right)}{dR^{2}}+\frac{1}{R}\frac{dU_{i}\left(R,\tau \right)}{dR}-\frac{U_{i}\left(R,\tau \right)}{R^{2}}-A\frac{1+\mu }{1-\mu }\frac{d\Theta_{i}\left(R,\tau \right)}{dR}=0$$
where U*_i_*(*R*,*τ*) is the dimensionless radial displacement. A is the dimensionless coefficient of thermal expansion. *μ* is Poisson’s ratio.

Similarly, introducing the geometric equation and stress–strain relationship into the outer boundary condition, it can be obtained that:(15)$$\frac{Y}{\left(1+\mu \right)\left(1-2\mu \right)}\left[\left(1-\mu \right)\frac{dU_{N}\left(R_{N},\tau \right)}{dR}+\mu \frac{U_{N}\left(R_{N},\tau \right)}{R}\right]-\frac{AY}{1-2\mu }\Theta_{N}\left(R_{N},\tau \right)=Q_{b}$$
where Q*_b_* is the dimensionless pressure on the outer boundary. Y is the dimensionless elastic modulus.

To sum up, the transient heat conduction equation and equilibrium equation of the multilayer model are obtained, and the definite solution conditions are given. Next, the Laplace transform method and series solution technique of differential equations are employed to solve the governing equation and equilibrium equation.

(3)Laplace Transformation and series solution

According to the RABT fire curve, the solution of the time-dependent temperature can be divided into three cases. Firstly, the temperature solution Θ^1^(*R*,*τ*) for 0 ≤ *τ* < *τ*_1_ is derived.

Using Laplace transformation, the time-dependent temperature Θ^1^(*R*,*τ*) can be transferred to [38,39]:(16)$$X_{i}\left(R,s\right)=\int_{0}^{\infty }\Theta^{1}{}_{i}\left(R,\tau \right)e^{-s\tau }d\tau$$
where Θ^1^*_i_*(*R*,*τ*) is the solution of the tunnel lining temperature for the rising stage (0 ≤ *τ* < *τ*_1_). X*_i_*(*R*,*s*) is the Laplace transformation of Θ^1^*_i_*(*R*,*τ*). *s* is the variable corresponding to the time *τ*.

Substitution of Equation (16) into Equation (2) gives:(17)$$\frac{\partial^{2}X_{i}\left(R,s\right)}{\partial R^{2}}+\frac{1}{R}\frac{\partial X_{i}\left(R,s\right)}{\partial R}=\frac{s}{K}X_{i}\left(R,s\right)$$

According to the series solution method of ordinary differential equations, the solution of Equation (17) could be expanded as the form of Fourier series:(18)$$X_{i}(R,s)=\sum_{n=0}^{\infty }D_{i,n}\left(s\right)\cdot {\left(R-1\right)}^{n}$$
where *D_i_*_,*n*_(*s*) is the coefficient. Substitution of Equation (18) into Equation (17) yields:(19)$$\left(n+1\right)\left(n+2\right)D_{i,n+2}\left(s\right)=-{\left(n+1\right)}^{2}D_{i,n+1}\left(s\right)+\frac{s}{K}\left[D_{i,n}\left(s\right)+D_{i,n-1}\left(s\right)\right]$$

*D_i_*_,*n*_ is the iterative formula of *D_i_*_,0_ and *D_i_*_,1_. It can be obtained under the initial boundary condition and continuity conditions. Next, by substitution of *D_i_*_,*n*_ in Equation (19), it can be expressed as:(20)$$X_{i}\left(R,s\right)=\sum_{n=0}^{\infty }\frac{F_{i,n}(s)}{W_{i,n}\left(s\right)}{\left(R-1\right)}^{n}$$

The detailed derivation is shown in Appendix B.

By using the inverse Laplace transform theory, the temperature solution Θ^1^(*R*,*τ*) is given:(21)$$\Theta^{1}(R,\tau )=\sum_{n=0}^{\infty }\Phi^{1}{}_{n}\left(\tau \right)\cdot {\left(R-1\right)}^{n}$$
where $\begin{array}{ll} \Phi^{1}{}_{n}= & \sum_{k=1}^{m}\left[\frac{1}{\left(m-k\right)!}\underset{s\to s_{1}}{\lim}\frac{d^{k-1}}{ds^{k-1}}\left[{\left(s-s_{1}\right)}^{m}\frac{F_{i,n}\left(s\right)}{W_{i,n}\left(s\right)}\right]e^{s\tau }\tau^{m-k}\right]{\left(R-1\right)}^{n}\\ & +\sum_{k=m+1}^{K}\frac{F_{i,n}\left(s_{k}\right)}{{\left[dW_{i,n}\left(s\right)/ds\right]|}_{s=s_{k}}}e^{s_{k}\tau }{\left(R-1\right)}^{n} \end{array}$.

To sum up, the time-dependent temperature for the rising stage is obtained. According to the results of previous research [38,39], the temperature solution for the holding and cooling stage can be achieved based on the solution of the temperature rising stage.

Similarly, by using the series solving method for ordinary differential equations, the solution of the equilibrium Equation (14) can also be expressed as Taylor’s series at *R* = 1:(22)$$U_{i}(R,\tau )=\sum_{n=0}^{\infty }B_{i,}{}_{n}\left(\tau \right)\cdot {\left(R-1\right)}^{n}$$
where *B_i_*_,*n*_(*τ*) is an unknown coefficient. It can be expressed as the following form:(23)$$\begin{array}{ll} B_{i,n+2} & =\frac{-1}{\left(n+1\right)\left(n+2\right)}[\left(n+1\right)\left(2n+1\right)B_{i,n+1}+\left(n^{2}-1\right)B_{i,n}\\ & -A\frac{1+\mu }{1-\mu }\left(n-1\right)\Phi_{i,n-1}-2A\frac{1+\mu }{1-\mu }\left(n\right)\Phi_{i,n}-A\frac{1+\mu }{1-\mu }\left(n+1\right)\Phi_{i,n+1}] \end{array}$$
where *B_i_*_,−1_ = Φ*_i_*_,−1_ = Φ*_i_*_,−2_ = 0.

The solutions of coefficients B0 and B1 can be obtained by solving the equilibrium Equation (14) under the boundary condition of Equation (15). Then, the solution of the displacement field U(*R*,*τ*) is solved by substituting *B_n_* into Equation (22). Finally, by using the geometric equation and the physical equation, the radial and circumferential stresses of the tunnel lining can be obtained. Above all, the general analytical solution for the transient temperature, displacement and thermal stress fields of the tunnel lining are all obtained based on the above theoretical derivation.

## 5. Discussions

### 5.1. Model Validation

The physical meanings and the values of the parameters used in the analytical model are listed in Table 4. The heat transfer coefficient *h*_2_ is determined as 8.33 W/m^2^K, which is the heat transfer coefficient of concrete at normal atmospheric temperature. The heat transfer coefficient h_1_ is greatly affected by fire temperature. As shown in Figure 11, the heat transfer coefficient h_1_ is very large at the beginning of fire, which is 500 W/m^2^K [35]. It decreased to less than 100 W/m^2^K after 5 min and gradually stabilized. It should be noted that the analytical solution will become very complex and difficult to solve when using the *h*_1_ in Figure 11. In order to simplify the calculation, the approximate treatment method is adopted to describe the heat transfer coefficient *h*_1_ in the form of the piecewise function. The piecewise function can be comprehensively determined according to the spalling time.

The existence of the measuring hole will lead to the leakage of water vapor, resulting in the reduction of pore pressure. Therefore, the setting of monitoring holes has a great impact on the test results. In the following analysis, only two sections are measured: the heated surface and the section 20 cm away from it. The reason is when the monitoring hole is far from the heated surface, it has little effect on the migration of water vapor and steam pressure near the heating surface.

Figure 12 compares the theoretical solution and the experiment temperature inside the concrete sample under fire. In both cases (maximum temperatures of 1000 °C and 800 °C), the spalling time obtained from the theoretical analysis was close to the experimental results, as well as the temperature history and distribution of the lining section with a depth of 20 cm from the heated surface. A reasonable agreement could be found from the comparison between the theoretical analysis and experimental results.

It can be found that the spalling time was very close in both cases (Figure 12a,b). The reason was that the size and material parameters of the samples are the same, especially the heating rate of fire. Spalling occurred before the fire temperature achieved the maximum value.

### 5.2. Temperature Field and Stress Field

Based on the proposed model, the temperature evolution of tunnel lining at any position could be obtained, as well as the thermal stresses. The creation of the research model was a challenge and made it possible to obtain very valuable information necessary for design. In the following analysis, the temperature and thermal stresses of the lining are discussed through a tunnel case. The inner and outer radii of the lining are set to 2.75 m and 3.1 m, respectively. The parameters still use the data provided in Table 4. Dimensionless temperature, radial stress and circumferential stress of the tunnel lining are illustrated in Figure 13 and Figure 14 at the moment of the first spalling. The highest temperature of the concrete lining is located at the heated surface. It decreases rapidly with the increase in depth from the heated surface. The circumferential stress is far greater than the radial stress (in Figure 14). The circumferential stress and the radial stress near the heated surface are compressive stress.

At the spalling moment, the temperature of the heated surface is 600 °C. *T*_1_ is 368 °C, which is the temperature of the spalling section away from the heated surface of 2 cm. Σ*_θ_*_1_ is 0.3784; that is, the circumferential stress of the spalling section is 113.5 MP. It is much larger than its compressive strength. Based on the model analysis, the circumferential stress of the section away from the heated surface of 5 cm is about 40 MP, which is approximately equal to its compressive strength. In other words, the circumferential stress is greater than its compressive strength for the section within 5 cm from the heated surface. However, the location of spalling is the section away from the heated surface of 2 cm. It can be concluded that in addition to the temperature gradient and the thermal stresses, the spalling of concrete under fire is also affected by other effects, such as the pore pressure. According to the analysis in Section 3.2 and Section 5.2, the critical temperature can be defined as that satisfying both the temperature of the heated surface is over 600 °C and the thickness of the spalling layer is 2 cm.

### 5.3. Spalling Depth

Figure 15 shows the difference in temperature field of tunnel lining with and without considering the concrete spalling under fire. The temperature of the residual part of the tunnel lining for considering the spalling is significantly higher than that without considering spalling. With the increase in elapsed time, the difference becomes more and more obvious. It also illustrates that it is necessary to consider the effect of spalling on the prediction of fire-induced damage in a tunnel.

Figure 16 presents the spalling depth evolution of tunnel lining with the elapse time increase. The maximum temperature of fire is 800 °C. The spalling depth increases nonlinearly with time. Before entering the cooling stage, a total of three spalling failures occurred. Based on the model proposed in this paper, the temperature and the stresses induced by fire could be obtained at every time. This can provide a theoretical basis for the damage evaluation of tunnel lining under fire.

## 6. Conclusions

The spalling characteristics of concrete under a unidirectional heating system are investigated in this study, with emphasis on the critical condition of concrete spalling under tunnel fire, which has a rapid heating rate, high peak temperature and long duration. Based on the experimental analysis, an analytical model considering the spalling of secondary concrete lining under tunnel fire is established to analyze the temperature and the thermal stress distribution of tunnel lining and obtain the sensitivity of spalling depth to heating time. The main conclusions are summarized as follows:(1)The spalling characteristics of concrete under tunnel fire are summarized from the heating tests. Based on the experimental test results, the moisture content of concrete is one of the key factors of spalling. Obvious layered spalling characteristics of concrete samples without drying could be observed under the unidirectional heat conduction system. The critical temperature of spalling is 600 °C, and the thickness of the spalling layer is 2 cm~2.5 cm.(2)Based on the spalling characteristics, a multilayer model considering the spalling of concrete lining under tunnel fire is established. Time-dependent temperature, stresses, and spalling depth of tunnel lining relating to the fire temperature could be obtained by using the model proposed. This is the basis of the damage assessment of tunnel lining after fire.(3)The spalling depth increases nonlinearly with time. Before entering the cooling stage, a total of three spalling failures occurred. The temperature and the stresses induced by fire could be obtained at every time based on the model proposed in this paper.(4)It is thus clear that heat transfer analysis without considering the effects of spalling can be erroneous at high temperatures. It is therefore imperative that the estimation of the fire resistance of a secondary concrete lining at high temperatures needs to consider its spalling and cross-sectional loss.

Before using this model for calculation and analysis, two key parameters must be determined, which are the critical temperature and the thickness of the spalling layer. Both parameters are difficult to determine and are related to the physical and mechanical properties of concrete materials, as well as the heating rate, the heating time and the maximum temperature. Therefore, a series of fire tests need to be carried out in the following research, to obtain the correlation between the spalling conditions and the mechanical parameters of concrete. This is the theoretical basis for quickly determining the spalling critical temperature and spalling layer thickness.

## Figures and Tables

**Figure 1 materials-15-03131-f001:**
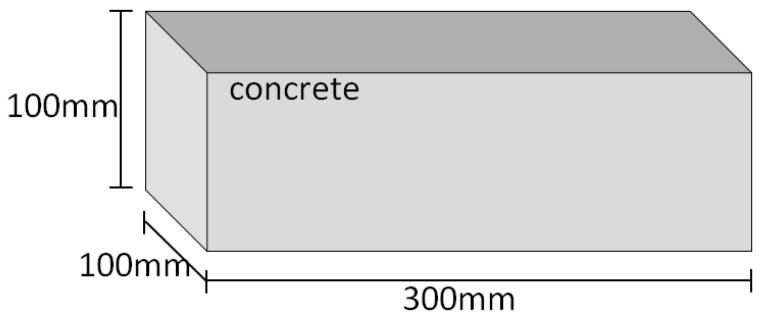
Concrete specimens.

**Figure 2 materials-15-03131-f002:**
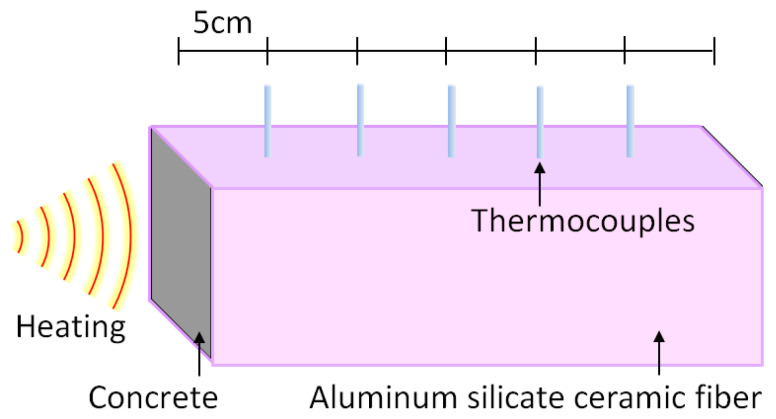
Heat treatment of concrete specimens.

**Figure 3 materials-15-03131-f003:**
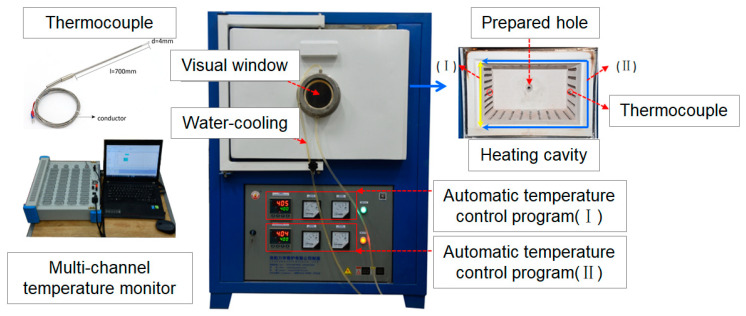
Muffle furnace and unidirectional heat transfer system.

**Figure 4 materials-15-03131-f004:**
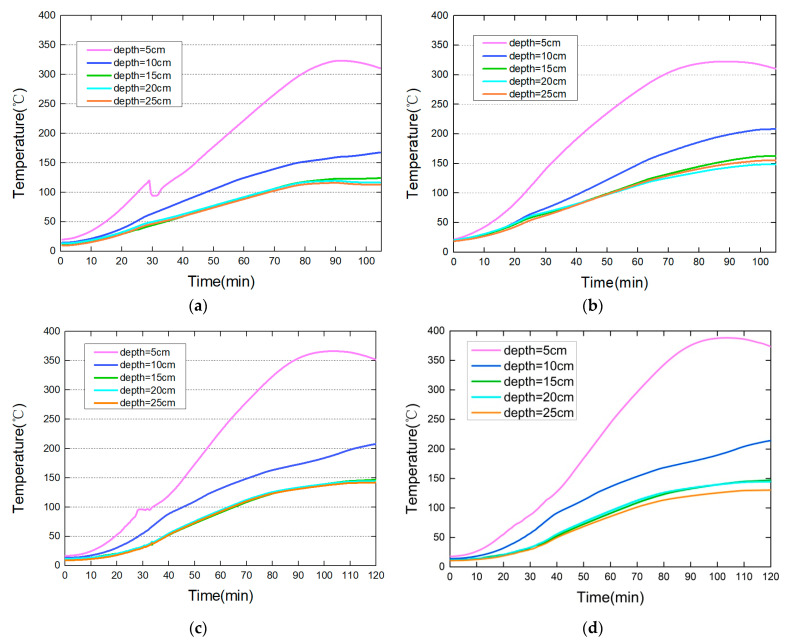
Temperature development inside the concrete samples: (**a**) 800 °C, samples without drying; (**b**) 800 °C, samples with drying at 105 °C for 36 h; (**c**) 1000 °C, samples without drying; (**d**) 1000 °C, samples with drying at 105 °C for 36 h.

**Figure 5 materials-15-03131-f005:**
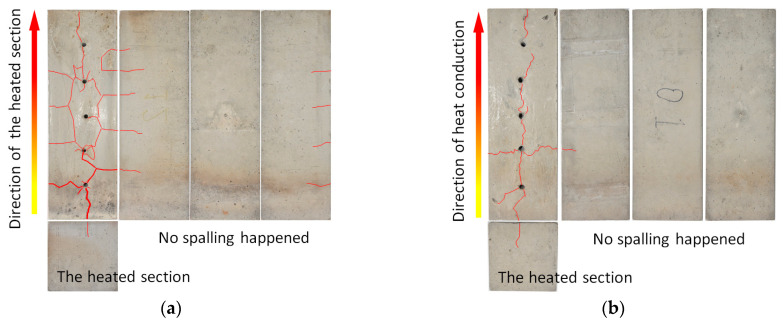
Results of fire tests with monitoring the internal temperature: (**a**) 800 °C, samples without drying; (**b**) 800 °C, samples with drying at 105 °C for 36 h.

**Figure 6 materials-15-03131-f006:**
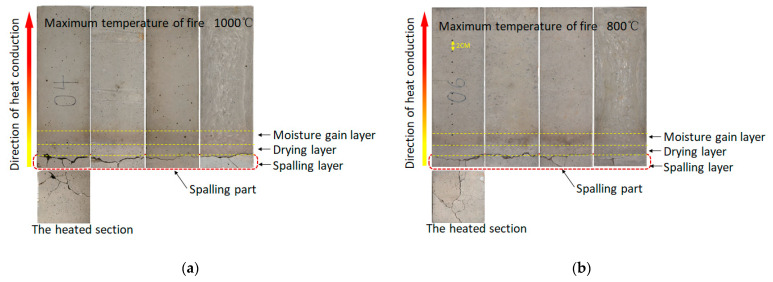
Images of the side surface of concrete sample under different fire temperatures: (**a**) maximum temperature of 1000 °C; (**b**) maximum temperature of 800 °C.

**Figure 7 materials-15-03131-f007:**
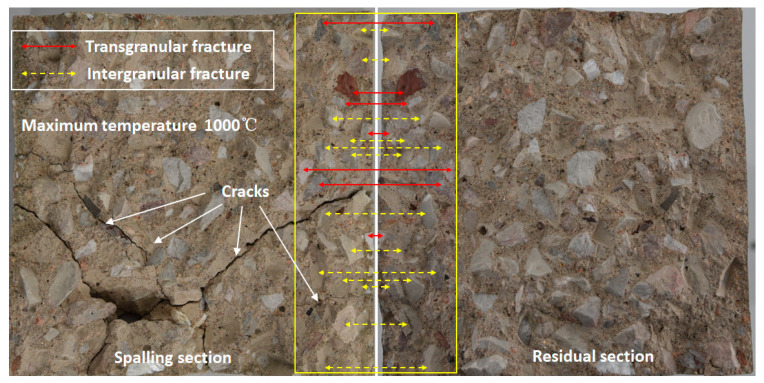
Morphology, structure and composition of fracture surface. (The back of the spalling section is the heated surface).

**Figure 8 materials-15-03131-f008:**
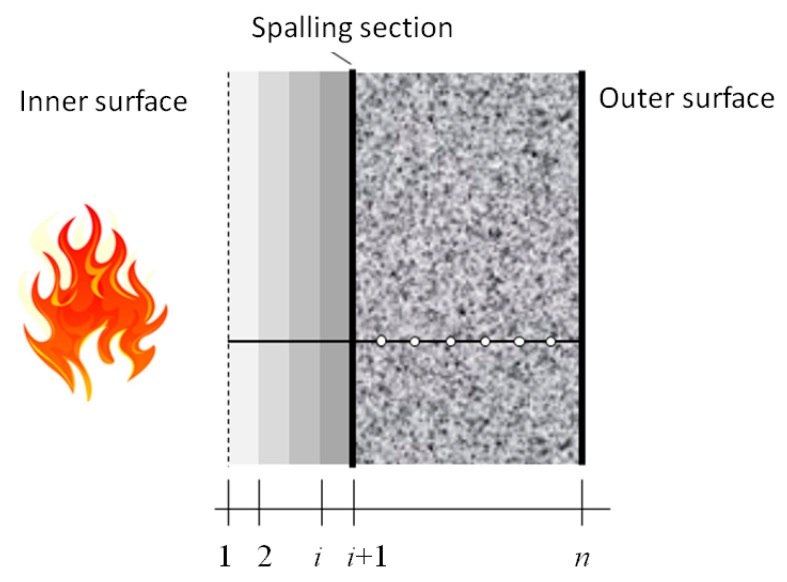
Schematic of the spalling process of concrete lining under tunnel fire.

**Figure 9 materials-15-03131-f009:**
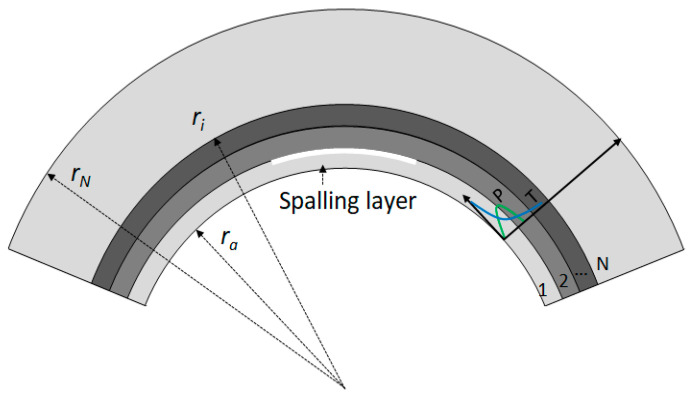
Layered spalling model.

**Figure 10 materials-15-03131-f010:**
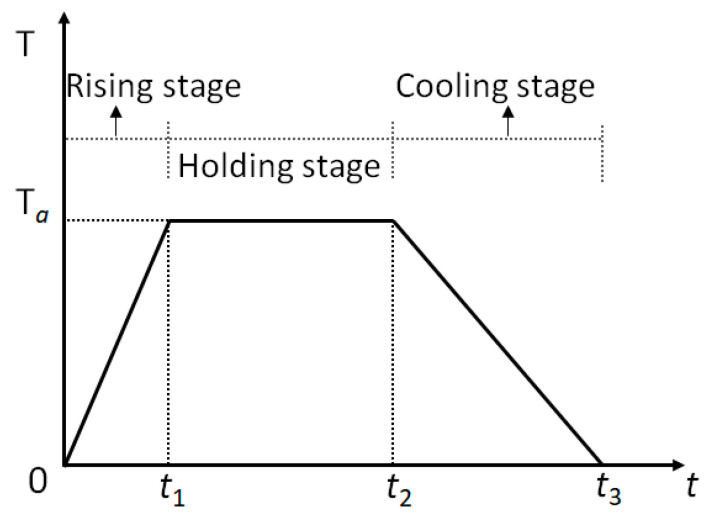
Employed temperature loading on the inner surface.

**Figure 11 materials-15-03131-f011:**
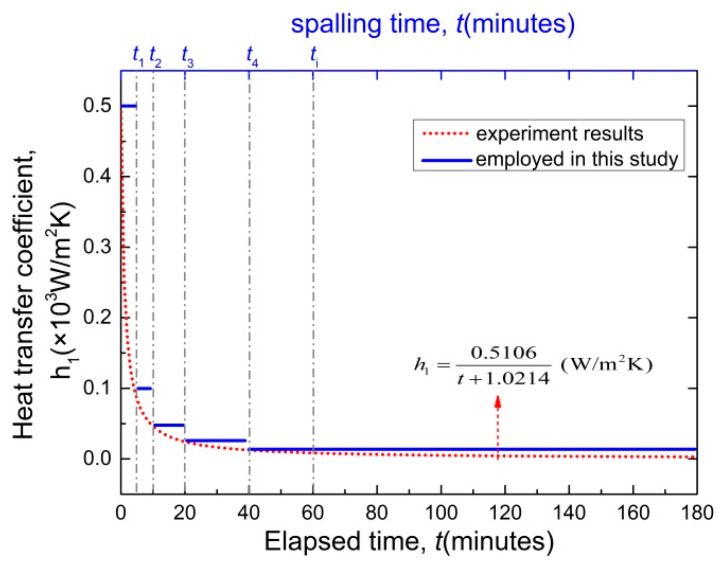
Heat transfer coefficient employed in this analysis.

**Figure 12 materials-15-03131-f012:**
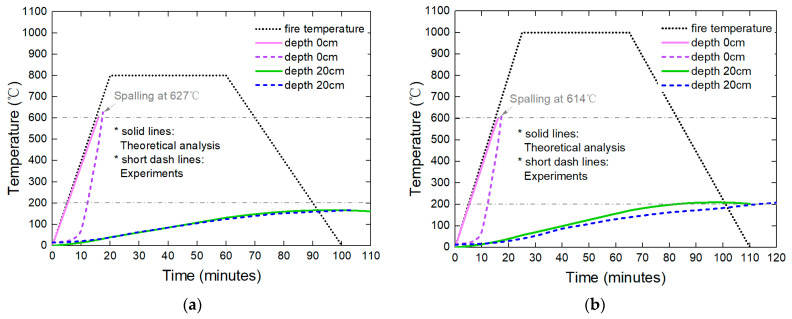
Comparison between theoretical analysis and the experiment results: (**a**) 800 °C; (**b**) 1000 °C.

**Figure 13 materials-15-03131-f013:**
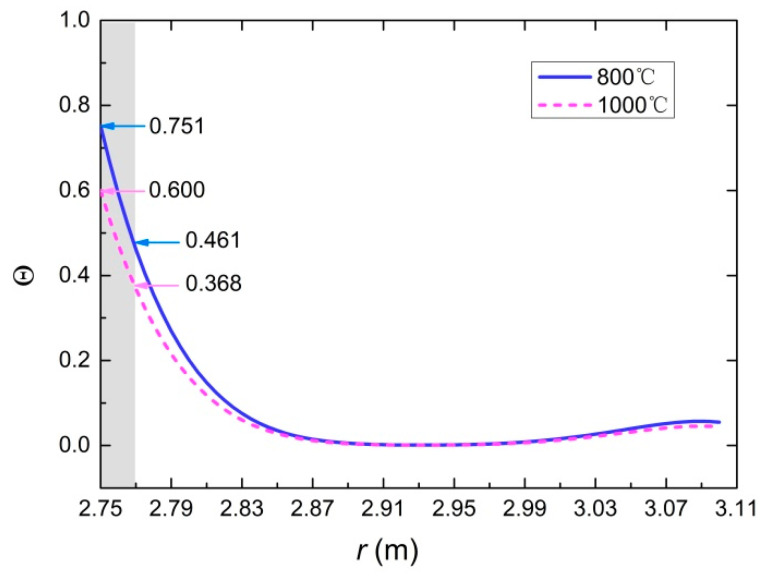
Dimensionless temperature distribution of tunnel lining at the spalling moment.

**Figure 14 materials-15-03131-f014:**
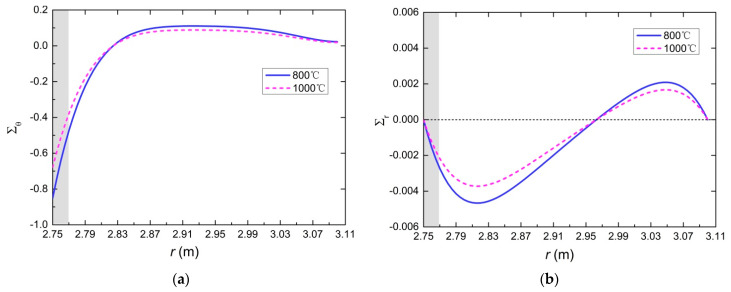
Dimensionless radial stress and circumferential stress at the spalling moment: (**a**) dimensionless circumferential stress; (**b**) dimensionless radial stress.

**Figure 15 materials-15-03131-f015:**
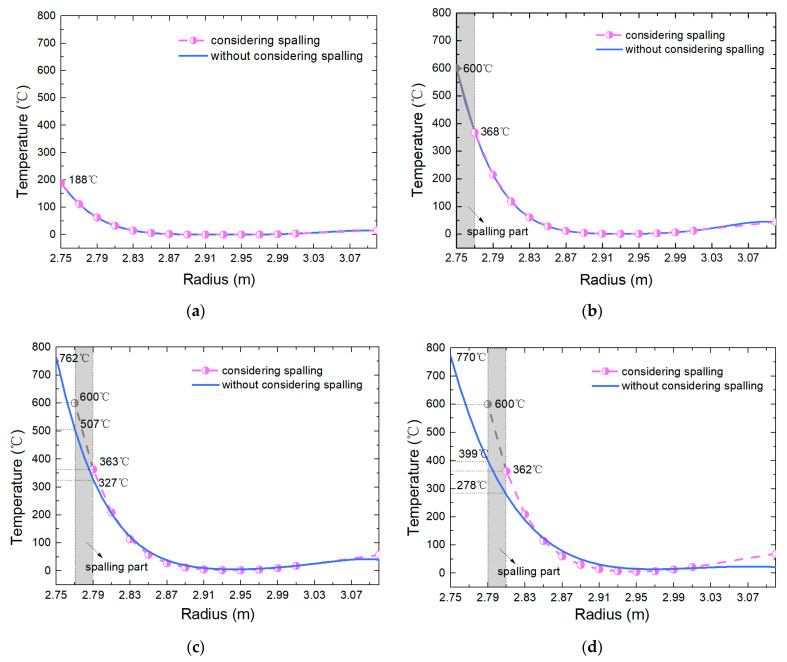
Temperature distributions of tunnel lining under fire (the maximum temperature of 800 °C) (**a**) 1 min after fire; (**b**) 1st spalling (15.88 min); (**c**) 2nd spalling (33.88 min); (**d**) 3rd spalling (56.82 min).

**Figure 16 materials-15-03131-f016:**
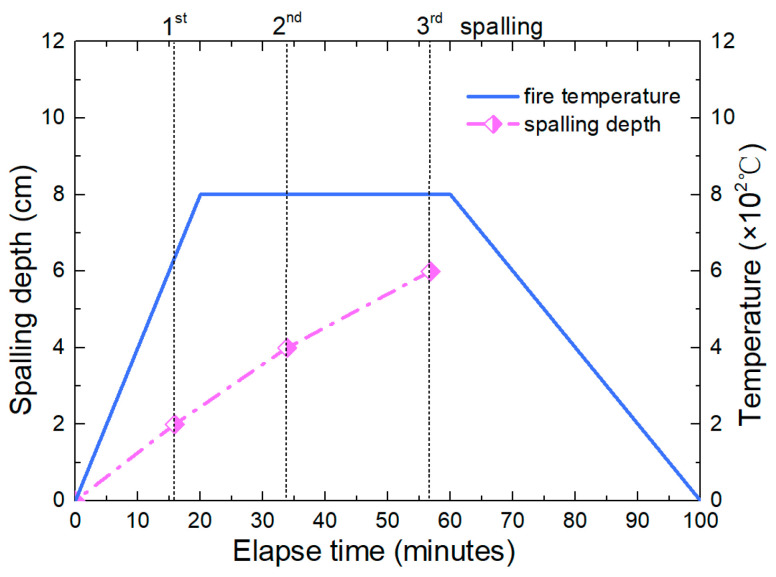
Spalling depth of tunnel lining with elapsed time.

**Table 1 materials-15-03131-t001:** Mixture proportions of concrete.

Concrete	w/c Ratio	Mass (kg/m^3^)						
		Cement	Water	Sand	Aggregate	Fly Ash	Expansive Agent	Admixture
C40P8	0.427	331	183	734	1057	83	22	7.8

**Table 2 materials-15-03131-t002:** Number of specimens.

	Number			
Specimen	Maximum Temperature of 800 °C		Maximum Temperature of 1000 °C	
	Without Drying	With Drying	Without Drying	With Drying
5 monitoring holes in specimen	3	3	3	3
0 monitoring hole in specimen	3	3	3	3
1 monitoring hole in specimen	3	0	3	0

Note: specimens were dried at 105 °C for 36 h before the heating tests.

**Table 3 materials-15-03131-t003:** Heating parameters.

	Heating Rate	Maximum Temperature	Holding Time	Cooling Mode
Case 1	40 °C/min	800 °C	30 min	Natural cooling
Case 2	40 °C/min	1000 °C	30 min	Natural cooling

**Table 4 materials-15-03131-t004:** Detailed mix proportion design of concrete lining.

Parameter	Temperature Dependence	In Put (at 20 °C)
Elastic modulus, E (GPa)	/	30
Poisson’s ratio, μ	/	0.2
Thermal expansion coefficient, α (/K)	/	1 × 10^−5^
Density, *ρ* (Kg/m^3^)	/	2400
Thermal conductivity coefficient, λ (W/(m·K))	$2-0.24\frac{T}{120}+0.012{\left(\frac{T}{120}\right)}^{2}$	1.883 [40]
Specific heat capacity, *c* (J/(kg·K))	836.8 + 0.4922 T	913 [40]

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
