# Peer review of "Explosive Spalling Mechanism and Modeling of Concrete Lining Exposed to Fire"

_materials, 2022, doi:10.3390/ma15093131_

Round 1
Reviewer 1 Report
This paper aims to establish an analytical model to estimate the influence of the concrete spalling on the fire damage depth prediction. The content of the article is comprehensive, but some details are not well done. There are some comments below that authors may consider revising their paper.
- The format of the paper needs to be changed according to the journal's requirements. Be sure to add line numbers!
- It is best to briefly summarize the experimental results in the abstract, which will be more attractive to the reader.
- There is no space in the introduction and conclusion's first paragraph.
- The last paragraph in the introduction should be revised to add what did this study do? Why did you do this? What are the advantages of doing this compared to previous research? What did this result?
- How is the average moisture content of 2.1% measured? Please add a description.
- Why is the water-cement ratio is 0.427 in Table 1?
- There is no period after the description sentence of the figure.
- The heating rate studied is 40℃, it took only 20 minutes to reach 800℃, and the maximum temperature hold time is 30 minutes. In Figure 4, why is the temperature still rising after 50 minutes?
- Positions 1, 2, 3 need to be identified in Figure 8.
Author Response
Dear Reviewers:
Thank you for your letter and for the reviewers’ comments concerning our manuscript entitled: “Explosive spalling mechanism and modeling of concrete lining exposed to fire” (materials-1657920). Those comments are all valuable and very helpful for revising and improving our paper. We have studied comments carefully and have made correction which we hope meet with approval. Revised portion is marked in red in the revised manuscript.

Reviewer 2 Report
The manuscript is well-written. However, the following comments need to be addressed first:
1. The abstract section needs to be re-written. The problem statement, results and benefits of the developed model should be added to it.
2. The introduction section needs to be separated from the literature review section. The problem statement and research objectives should be included in the introduction section.
3. More recent studies need to be added in the literature review section such as:
Hua, N., Khorasani, N. E., Tessari, A., & Ranade, R. (2022). Experimental study of fire damage to reinforced concrete tunnel slabs. Fire Safety Journal, 127, 103504.
Al-Sakkaf, A., Mohammed Abdelkader, E., Mahmoud, S., & Bagchi, A. (2021). Studying Energy Performance and Thermal Comfort Conditions in Heritage Buildings: A Case Study of Murabba Palace. Sustainability, 13(21), 12250.
Cheng, P., Zhu, H., Zhang, Y., Jiao, Y., & Fish, J. (2022). Coupled thermo-hydro-mechanical-phase field modeling for fire-induced spalling in concrete. Computer Methods in Applied Mechanics and Engineering, 389, 114327.
4. It is not clear what are the distinctive features of the proposed optimal field sampling method.
5. Limitations of previous studies should be added to the manuscript.
6. A research framework figure and section should be present to show the steps of the developed model.
7. Model validation and results need to be collected in a separate section towards the end of the manuscript.
8. The conclusion section should be strengthened. More insight into the result should be added.
9. Limitations of the present research study should be added at the end of the conclusion section.
10. The existing format is not MDPI's required format, the authors should put the manuscript in the right format.
Author Response
Dear Ms. Alice Gu, and Reviewers:
Thank you for your letter and for the reviewers’ comments concerning our manuscript entitled: “Explosive spalling mechanism and modeling of concrete lining exposed to fire” (materials-1657920). Those comments are all valuable and very helpful for revising and improving our paper. We have studied comments carefully and have made correction which we hope meet with approval. Revised portion is marked in red in the revised manuscript.

Reviewer 3 Report
This paper examines explosive spalling mechanism and modeling of concrete lininig exposed to fire.
Line 113 After 28days of standard curing at a temperature of 26℃ - in accordance with which recommendations
Line 114 - the samples were left aging and drying in a laboratory for one year. (humidity, temperature)
Lines 116-118 - whether mi and m0 were measured on the same samples?
Line 130 - Why was an expansive agent added to the mixture? What does admixture mean?
If I understood correctly, there are 2 series of specimens: wrapped and unwrapped? How many specimens are in each series? Did both series have thermocouples?
Line 136 - The diameter of measuring hole is 4cm. Diameter or depth?
Line 163 - The duration of the maximum temperature was 30mins. Why? It is usually from 1 to 2.5 hours.
There are heating conditions in accordance with the recommendations of the RILEM commission for testing the compressive strength:
- RILEM TC 129 MHT: Test methods for mechanical properties of concrete at high temperatures – Part 3: Compressive strength for service and accident conditions. Materials and Structures, 28, 181, 1995, pp. 410–414.
- RILEM TC 200 HTC: Mechanical concrete properties at high temperatures – modelling and applications – Part 1: Introduction, General presentation. Materials and Structures, 40, 2007, pp. 841–853.
In accordance with what recommendations the authors decided on heating parameters?
Natural cooling - in the oven? In the oven with the door open? Out of the oven?
Spalling phenomenon - what other authors say about the causes
Lines 194-197 - Compare with the works of other authors who have shown the dependence of the temperature distribution on the cross section depending on the duration of ISO fire curves.
Line 268 (Figure 7) Mark heated surface
Line 422 - whether the parameters of Table 3 were measured on the test concrete?
Line 442 – Figure 12 - Please, show experimental and theoretical curves for all measuring depths
Line 482 – Figure 15 – 15.88mins? – minutes?
How much does the composition of concrete affect the obtained results?
Can spalling occur at lower but longer temperatures?
Round 2
Reviewer 1 Report
The authors revised this work.
Author Response
Once again, thank you very much for your comments and suggestions.
Reviewer 3 Report
- Line 442 – Figure 12 - Please, show experimental and theoretical curves for all measuring depths
Response: Thank you very much. In this section, only two sections were monitored: one was the heated surface and the other was the section of 20cm away from the heated surface. Why choose the section of 20cm away from the fire surface? The reason is that when the monitoring hole is close to the heated surface, it will provide an escape channel for the vapor caused by the high temperature, results in the decrease of vapor pressure. The existence of monitoring hole seriously affects the result of spalling. Therefore, monitoring holes cannot be set near the heated surface. When the distance is long, this effect is significantly reduced.
Yes, but the greater the distance, the better the experimental and theoretical curves match, and we do not know the real situation in the critical zone.